# Synthesis, Biological Activity, ADME and Molecular Docking Studies of Novel Ursolic Acid Derivatives as Potent Anticancer Agents

**DOI:** 10.3390/ijms24108875

**Published:** 2023-05-17

**Authors:** Olga Michalak, Marcin Cybulski, Wojciech Szymanowski, Agnieszka Gornowicz, Marek Kubiszewski, Kinga Ostrowska, Piotr Krzeczyński, Krzysztof Bielawski, Bartosz Trzaskowski, Anna Bielawska

**Affiliations:** 1Chemistry Section, Pharmacy, Cosmetic Chemistry and Biotechnology Research Group, Łukasiewicz Research Network–Industrial Chemistry Institute, 8 Rydygiera Str., 01-793 Warsaw, Poland; marcin.cybulski@ichp.lukasiewicz.gov.pl (M.C.); piotr.krzeczynski@ichp.lukasiewicz.gov.pl (P.K.); 2Department of Biotechnology, Medical University of Bialystok, 1 Kilińskiego Str., 15-089 Bialystok, Poland; wojtekszymanowski@wp.pl (W.S.); agnieszka.gornowicz@umb.edu.pl (A.G.); anna.bielawska@umb.edu.pl (A.B.); 3Analytical Research Section, Pharmaceutical Analysis Laboratory, Łukasiewicz Research Network–Industrial Chemistry Institute, 8 Rydygiera Str., 01-793 Warsaw, Poland; marek.kubiszewski@ichp.lukasiewicz.gov.pl; 4Department of Organic Chemistry, Faculty of Pharmacy, Medical University of Warsaw, 1 Banacha Str., 02-097 Warsaw, Poland; kostrowska@wum.edu.pl; 5Department of Synthesis and Technology of Drugs, Faculty of Pharmacy, Medical University of Bialystok, 1 Kilińskiego Str., 15-089 Bialystok, Poland; krzysztof.bielawski@umb.edu.pl; 6Chemical and Biological Systems Simulation Lab, Center of New Technologies, University of Warsaw, 2C Banacha Str., 02-097 Warsaw, Poland; b.trzaskowski@cent.uw.edu.pl

**Keywords:** amino acid, antitumor, cancer, docking, ursolic acid

## Abstract

A series of new ursolic acid (UA) derivatives substituted with various amino acids (AAs) or dipeptides (DP) at the C-3 position of the steroid skeleton was designed and synthesized. The compounds were obtained by the esterification of UA with the corresponding AAs. The cytotoxic activity of the synthesized conjugates was determined using the hormone-dependent breast cancer cell line MCF-7 and the triple-negative breast cancer cell line MDA. Three derivatives (*l*-seryloxy-, *l*-prolyloxy- and *l*-alanyl-*l*-isoleucyloxy-) showed micromolar IC_50_ values and reduced the concentrations of matrix metalloproteinases 2 and 9. Further studies revealed that for two compounds (*l*-seryloxy- and *l*-alanyl-*l*-isoleucyloxy-), a possible mechanism of their antiproliferative action is the activation of caspase-7 and the proapoptotic Bax protein in the apoptotic pathway. The third compound (*l*-prolyloxy- derivative) showed a different mechanism of action as it induced autophagy as measured by an increase in the concentrations of three autophagy markers: LC3A, LC3B, and beclin-1. This derivative also showed statistically significant inhibition of the proinflammatory cytokines TNF-α and IL-6. Finally, for all synthesized compounds, we computationally predicted their ADME properties as well as performed molecular docking to the estrogen receptor to assess their potential for further development as anticancer agents.

## 1. Introduction

Breast cancer (BC) is an important global health problem [1] and one of the most common types of cancer among women. Over 2.3 million new cases and 685,000 deaths from breast cancer occurred in 2020 with geographic variation across countries and world regions. The incidence rates range from <40 per 100,000 females in some Asian and African countries to over 80 per 100,000 in Australia/New Zealand, Northern America, and parts of Europe [2]. In Poland, in the years 2000–2019, breast cancer was diagnosed in 315,278 patients (the ratio of men to women was 1/100) [3]. BC metastasis is responsible for 90% of human cancer-related deaths. Thus, it remains one of the great impediments in curing cancer [4]. Many anticancer drugs in use today are nonspecific to cancer cells and, therefore, normal cells are also destroyed, resulting in severe side effects, such as secondary (therapeutic-induced) malignancies, nephron-, hepato-, neuro-, cardio-, and ototoxicity [5]. For this reason, the development of new drugs is imperative, and research is ongoing to understand the difference between healthy cells and cancer ones. Some of the differences relate to the presence of increased activity of certain enzymes or overexpression of receptors located on the surface of tumor cells [6]. For example, the overexpression of CES2, belonging to the largest class of enzymes in humans called serine hydrolases (SH), has been observed in pancreatic adenocarcinoma compared to paired noncancerous tissues [7]. The discovery of new chemical compounds that involve the activity of enzymes implied in cancer growth or influence the activity of receptors may lead to the development of new and effective chemotherapeutic agents.

One of the effective methods for obtaining new drugs is the chemical modification of naturally occurring substances with proven high biological activity. An example of such a compound is UA (3β-hydroxy-urs-12-en-28-oic acid), which belongs to natural compounds from the terpenes group. UA is a pentacyclic triterpenoid derived from berries, leaves, flowers, and fruits of medicinal plants (e.g., lavender, oregano, thyme, and marigold [8]), and it has been reported to possess a wide range of pharmacological properties, including antiallergic, antiviral, antiinflammatory, antibacterial, and antitumor activities [9,10,11]. Recent studies have shown that UA displayed in vitro antitumor effects and cytotoxic activity against various types of cancer cell lines [12,13,14]. It also induces apoptosis through intrinsic and extrinsic apoptotic pathways [15]. There are numerous data on the activity, detailed mechanism of action, and therapeutic effects of UA against various BC lines (Figure 1), making this pentacyclic triterpene a promising candidate for a therapeutic agent against BC [15,16,17,18]. For instance, A. Bishayee et al. showed in vitro and in vivo data suggesting that UA may be effective in the prevention and therapy of human BC by acting on MCF-7 cell proliferation throughout the cytostatic response in the G1 phase of the cell cycle followed by cell death [19]. UA has also been reported to inhibit the migration of T47D, MCF-7, and MDA-MB-231 BC cells by Bcl-2/Caspase-3 signaling pathways [20]. Lewińska et al. tested UA against phenotypically distinct BC cells MCF-7 (ER^+^, PR^+/−^, HER2^−^), MDA-MB-231 (ER^−^, PR^−^, HER2^−^), and SK-BR-3 (ER^−^, PR^−^, HER2^+^). Results revealed that UA caused G0/G1 cell cycle arrest, which was accompanied by oxidative stress and DNA damage. UA was found to be a potent inducer of apoptosis, which was achieved by targeting the glycolytic pathway and autophagy in breast cancer cells [21]. The suppression of glycolytic metabolism was also demonstrated by Wang et al. [22] as the main cause of BC growth and metastasis inhibition. Cell proliferation induced by UA was observed on MCF-7 and MCF/ADR human BC cell lines and, interestingly, the adriamycin-resistant BC cell line showed no resistance to UA [23].

It was also shown to inhibit MCF-7 cell growth in a dose- and time-dependent manner by inhibiting the FoxM1protein expression and CyclinD_1_/CDK_4_ that induced apoptosis [24]. Yeh et al. [25] showed migration and metastasis suppression of MDA-MB-231 BC by modulation of c-Jun *N*-terminal kinase (JNK) and mammalian target of rapamycin signaling. In MDA-MB-231 human breast adenocarcinoma, UA increased the expression levels of the Fas receptor, caused caspase-8, -3, PARP cleavage, and down-regulated Bcl-2 [26,27], which were involved in the extrinsic death receptor pathway. Furthermore, it induced mitochondrial death by releasing cytochrome c from mitochondria and caspase-9 cleavage.

Due to the various biological activities of UA, it has been considered by selected research groups as the lead compound for chemical derivatization, to find more efficient potential therapeutics. Most researchers modified the molecular structure of UA at C-2 or hydroxyl (C-3) and carboxylic functional group (C-28) positions on the triterpenoid ring. The studies of V. Khwaza [28] defined that substitution at these three key sites led to an improvement in the biological activities of UA derivatives. The obtained compounds were tested in vitro against various cancer cell lines, including BC and it was shown that structural modification significantly improved the antiproliferative activity, when compared to the respective reference molecule.

**Figure 1 ijms-24-08875-f001:**
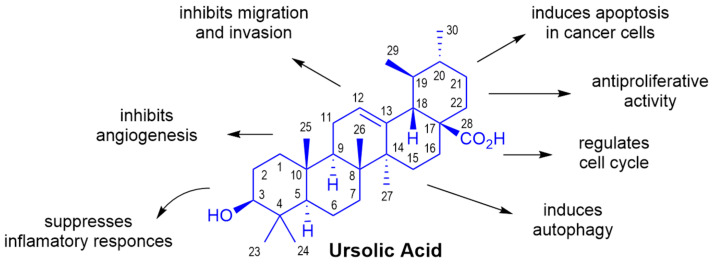
Effects of the ursolic acid’s action on breast cancer cells [17,20,29].

Wen Gu et al. [30] synthesized a series of new quinoline derivatives of UA and evaluated in vitro their cytotoxicity against three human cancer cell lines: MDA-MB-231, HeLa, and SMMC-7721. The most potent derivative showed IC_50_ values of: 0.61 ± 0.07, 0.36 ± 0.05, and 12.49 ± 0.08 µM against MDA-MB-231, HeLa, and SMMC-7721 cells, respectively, which were also higher than for the positive control (etoposide). Cell cycle analysis also indicated that this new compound could cause cell cycle arrest of MDA-MB-231 cells in the G0/G1 phase. On the other hand, A.S. Leal et al. [31] obtained a series of new UA fluorolactone derivatives, and 2-cyano-3-oxo-12α-fluoro-urs-1-en-13,28β-olide was found to be the most effective inhibitor of cancer cell growth. This compound also exhibited improved antiproliferative activities against MCF-7, PC-3, Hep G2, and A549 cancer cell lines with IC_50_ values lower than 1 µM.

Later, S. Rashid et al. [32] designed and synthesized the series of UA 1-phenyl-1*H*-[1,2,3]-triazol-4-ylmethyl ester derivatives, which were screened for anticancer activity against the panel of four human cancer cell lines including A-549, MCF-7, HCT-116, THP-1, and a normal epithelial cell line (FR-2). Pharmacological results showed that most of the compounds displayed a high level of antitumor activities against the tested cancer cell lines when compared to UA. Similarly, B. A. Dar et al. [33] obtained a library of UA benzylidine derivatives, which were evaluated against four human carcinoma cell lines including A-549, MCF-7, HCT-116, THP-1, and a normal human epithelial cell line. All compounds exhibited improved cytotoxicity against the tested carcinoma cell lines with respect to UA.

In 2013, D. Manciula et al. [34] synthesized three UA derivatives substituted with a DP (Ala-Gly-, Gly-Leu-, and Leu-Gly-) at the C-28 position. The antiproliferative test showed that there were no major differences between the activity of UA and its substituted derivatives for the A431 and HeLa cell lines. Wei Li et al. [1] synthesized novel derivatives of UA with a nitrogen-containing additional heterocyclic scaffold connected to the pentacyclic terpenoid A ring and with piperazine at the C-28 position. The compound blocked BC cell progression by inducing apoptosis and cell-cycle arrest in the S and G0/G1 phases and significantly repressed the proliferation of SUM149PT and HCC1937 cancer cell lines in a dose-dependent manner.

Modification in the carboxylic group by substitution of the piperazinylbutyl group led to a novel active compound UA232 with strong antitumor activity. In vitro experiments showed that UA232 inhibited the proliferation of MCF-7 through G0/G1 arrest and promoted apoptosis. Furthermore, UA232 suppressed tumor growth in a mouse xenograft [35]. Later, Nedopekina et al. [36] linked the lipophilic triphenylphosphonium cation to the C-28 position by the hydrophobic *n*-butyl or hydrophilic triethylene glycol spacers. In selected compounds, dichloroacetate, a known pyruvate dehydrogenase kinase (PDK) inhibitor, was introduced at the 3β-OH position. Compounds bearing the triphenylphosphonium group exhibited high cytotoxicity against MCF-7; although, their subsequent conjugation with dichloroacetate did not produce any synergistic cytotoxicity effect. Finally, the first examples of UA derivatives having a cationic scaffold derived from malachite green were designed and obtained. Derivatives with piperazinylamide spacer and substituted carboxylated malachite green analogs were cytotoxic for selected human tumor cell lines, including MCF-7 human breast adenocarcinoma cells [37].

Considering all the previous results for UA derivatives against BC cell lines, and their antiproliferative activity against breast adenocarcinoma, we designed, synthesized, and characterized a new family of UA derivatives substituted with various AAs and DP at the C-3 position. All analogs of UA were evaluated in vitro for their activity profile against cancer cell lines (ER, PR positive MCF-7, and triple-negative MDA-MB-231). The cytotoxicity of all compounds was determined by using the MTT assay. To demonstrate that all tested compounds exhibited antiproliferative activity, the metabolic incorporation of [^3^*H*]-thymidine into DNA was measured to monitor the rate of DNA synthesis. Additionally, the influence of the newly synthesized compounds on the induction of apoptosis and the activation of caspase-7 in human breast cancer cells was examined. Finally, the interaction of new compounds with proteins involved in autophagy processes, such as LC3A, LC3B, and Beclin-1, as well as their influence on the inflammatory response, was explored and molecular docking studies were used to elucidate their potential mechanism of action.

## 2. Results and Discussion

### 2.1. Chemistry

In this study, a series of UA derivatives has been synthesized, including seven ursoloil esters of selected AAs (**3c**, **4c**, **5a**–**9a**) and three DP derivatives (**10a**–**12a**), as shown below in Figure 1.

First, UA was converted to its methyl ester **2**. Compound **2** was then subjected to reaction with the corresponding protected *N*-*t*-butoxycarbonyl AAs in the presence of *N*,*N*′-dicyclohexylcarbodiimide (DCC) and 4-dimethylaminopyridine (DMAP) in dichloromethane (DCM) at room temperature. The crude products of coupling reactions were then purified by flash chromatography (ethyl acetate/hexane) to yield (46 to 81%) **3**–**9** intermediates. In the next step, Boc groups were removed by a 2.9 M solution of gaseous HCl in ethyl acetate. The deprotection gave the appropriate hydrochlorides **5a**–**9a** as final products with good yields.

DP derivatives **10a**–**12a** were synthesized using a “step-by-step” method [38]. The 3β-(*l*-isoleucyloxy)-urs-12-en-28-oic acid methyl ester hydrochloride **6** was coupled with Boc-*l*-Ala and with *O*-benzotriazol-1-yl-*N*,*N*,*N*’,*N*’-tetramethyluronium tetrafluoroborate (TBTU) as the coupling activator in the presence of 1-hydroxybenzotriazole (HOBt) and *N*,*N*-diisopropylethylamine (DIPEA) in DCM [39]. Likewise, 3β-(*l*-prolyloxy)-urs-12-en-28-oic acid methyl ester hydrochloride **7a** was coupled with Boc-*l-*Ile by the same protocol to afford **11**. Similarly, the coupling of **7a** with Boc-*l*-Leu under these conditions allowed a derivative **12** to be obtained. The crude products **10**–**12** were separated and purified by flash chromatography (ethyl acetate–hexane) to obtain **10**–**12** with good yields of 80%, 87%, and 81%, respectively. Then, Boc group removal with gaseous hydrogen chloride in ethyl acetate led to expected hydrochlorides **10a**–**12a**.

UA derivatives **3b** and **4b** were obtained after removal of the benzyl groups from the *N*-protected compounds **3** and **4** using the reduction method [40], i.e., catalytic hydrogenation with Pd/C in ethyl acetate. After the reaction had been completed, the mixtures were filtered through cellite pads to give, after evaporation of the filtrates, products **3b** and **4b,** with good yields. In the final step, Boc groups were removed by hydrogen chloride gas in ethyl acetate to form hydrochlorides **3c** and **4c** with yields of 65% and 75%, respectively. Upon synthesis, all compounds were characterized by ^1^H NMR, ^13^C NMR, and HRMS techniques.

### 2.2. Biological Studies

#### 2.2.1. Cell Viability and Proliferation in MCF-7 and MDA-MB-231 Breast Cancer Cells

The effect of new UA derivatives on the cell viability of MCF-7 and MDA-MB-231 human BC cell lines was evaluated using the MTT assay (Table 1). Cells were incubated with varying concentrations of tested compounds and reference one (UA) for 24 h to determine IC_50_ values. The tests were performed according to the manufacturer’s instructions (see Section 3.3.2 for details).

Based on the results obtained after 24 h incubation of cells with the tested compounds, we showed that compounds **4c**, **5a**, **7a**, **10a**, **11a**, and **12a** caused a significant reduction in the cell viability of human BC cell lines. These compounds were the most effective against triple-negative MDA-MB-231 cells with comparable IC_50_ values: 15 μM, 17.5 μM, 14 μM, 17 μM, 17.5 μM, and 17.5 μM and had slightly lower activity against estrogen-dependent MCF-7 cells (IC_50_: 32 μM for **4c**, 44 μM for **5a**, 42 μM for **7a**, 50 μM for **10a**, 81 μM for **11a,** and 70 μM for **12a**). UA with the IC_50_ value of 32.5 µM was not as effective in decreasing the viability of MDA-MB-231 cell lines when compared to its most active derivatives. Among the synthetized compounds, **3c** showed no activity while compounds **2** and **9a** exhibited high selectivity to MDA-MB-231 cells.

To investigate the effect of new UA derivatives on cell proliferation in MCF-7 and MDA-MB-231 cells, the level of [^3^*H*]-thymidine incorporation into the DNA of human BC cells was measured. The obtained results are shown in Table 2 and suggest that exposure of cancer cells to new UA derivatives inhibited cell proliferation in a concentration-dependent manner.

In general, MDA-MB-231 cells were more sensitive to the tested agents than MCF-7. Although, compound **10a** was the most effective in decreasing MDA-MB-231 cell proliferation with an IC_50_ value of 14.9 μM, compounds **4c**, **5a**, **7a**, **11a**, and **12a** were comparable effective against cell proliferation in this line, i.e., with the IC_50_ values lower than 20 μM. Based on these results we selected compounds **4c**, **7a,** and **10a** for more detailed biological studies.

#### 2.2.2. Proapoptotic Activity

Bax, a proapoptotic member of the Bcl-2 family, plays an important role in the neutralization of the antiapoptotic proteins Bcl-2, Bcl-xL, and Mcl-1. Bax, which is involved in the intrinsic mitochondrial pathway of apoptosis, promotes mitochondrial outer membrane permeabilization (MOMP) and the formation of a complex named apoptosome. The activation of the downstream executor caspases-3, -6, and -7 responsible for apoptotic cell death is also observed [41]. In our studies, the concentrations in MDA-MB-231 cells were measured after 24 h incubation with the tested compounds. The results shown in Figure 2A suggest that all tested compounds led to a statistically significant increase in Bax concentration. Compound **7a** increased the Bax level from 479 ng/mL in the control sample to 614.33 ng/mL. The effects of increasing Bax concentrations to 672.33 ng/mL and 585.68 ng/mL were also observed after administration of **4c** and **10a** compounds. All these values were lower than for UA (832.67 ng/mL), which demonstrated the highest increase in Bax concentration. Additionally, we also decided to confirm the ability of UA derivatives to induce apoptosis by measuring the level of caspase-7 after administration of the tested compounds (Figure 2B). The results show that compound **4c** significantly increased caspase-7 levels to 7.65 ng/mL, while other tested compounds led only to a slight increase in caspase-7 levels, but the values are not statistically significant.

#### 2.2.3. Autophagic Activity

Autophagy is a process still not completely understood, but known to play a role in cancer progression and metastasis [42]. The most important features of this process are associated with microtubule-associated proteins 1A/1B light chain 3A and 3B (LC3A and LC3B), as well as Beclin-1. The microtubule-associated protein 1A/1B light chain 3B protein, LC3B, is an adapter protein that directly interacts with numerous autophagy cargo receptors, initiation complexes, processing enzymes, adapters, and trafficking proteins [43]. In our study, LC3A, LC3B, and Beclin-1 concentrations were analyzed after 24 h of cell incubation with the newly synthesized compounds (Figure 3). Of all tested derivatives, only compound **7a** increased LC3A concentration (2.69 ng/mL) in comparison with the control (1.13 ng/mL), while the other four tested agents did not affect the LC3A concentration (Figure 3A). In the LC3B case, we also observed that only compound **7a** increased the concentration of autophagic marker from 0.57 ng/mL in untreated MDA-MB-231 cell lysates to 1.53 ng/mL, and the difference was statistically significant (Figure 3B). Finally, for Beclin-1 none of the compounds showed any increase in Beclin-1 concentration (Figure 3C).

#### 2.2.4. MMP Activity

Matrix metalloproteinase-2 and -9 are involved in the propagation of various types of neoplasms, including BC. Both proteins have been associated with the invasive and metastatic stages of BC [44,45,46]. A low level of MMP-2 is associated with a favorable prognosis in patients with a hormone receptor-negative tumor, usually associated with an elevated risk of death. As a result, blocking MMP-2 secretion and activation during breast carcinoma can decrease metastasis and increase survival rate [47]. In our study, the initial concentration level of MMP-2 was determined in control samples at 149.67 pg/mL. UA caused a significant increase in the concentration of MMP-2 to 342.67 pg/mL. The opposite effect was observed for the tested derivatives, as they maintained the same level of MMP-2 concentration as the control (Figure 4A). Additionally, compound **7a** turned out to be the most effective in reducing the level of MMP-2. The analyzed metalloproteinase in **7a** was determined at the level of 117.67 pg/mL. In the case of MMP-9, all compounds decreased the MMP-9 concentration in cell lysates. The most effective inhibitor was UA with MMP-9 concentration at 503.143 pg/mL (Figure 4B).

#### 2.2.5. Anti-Inflammatory Activity

TNF-α, similarly to IL-6, is a proinflammatory cytokine, characterized by a broad spectrum of functions that include cytotoxic and cytostatic effects against cancer cells [48]. Decreasing its activity (e.g., with specific inhibitors) is used in the treatment of a number of autoimmune diseases and diseases related to chronic, destructive inflammation [49]. In the case of our new derivatives, all of them reduced both IL-6 and TNF-α concentrations in MDA-MB-231 cells (Figure 5). In the first case, the most effective compound was **7a,** reducing the control level of 454 pg/mL to 271 pg/mL; in this case, the pure UA increased IL-6 levels with respect to the untreated control (Figure 5A). In the latter case, only **4c** and **7a** significantly decreased TNF-α concentration, with **4c** being the more effective compound (Figure 5B).

### 2.3. ADME Prediction

The major predicted ADME properties of all studied compounds were presented in Table 3. Almost all new compounds exceed the commonly desired limit of 500 Da for systems with good oral bioavailability [50], but fall within the modern limit of <700 Da [51]. In general, all estimated properties fell within the extended Lipinski’s rule of five spaces with some rare exceptions (such as the log *p* value > 8) that were only slightly beyond this rule [52]. As in the case of the diosgenin and tigogenin conjugates studied previously, the only true point of concern at this stage may be the relatively high lipophilicities of all UA derivatives, resulting from the high hydrophobicity of the parent system. In fact, UA is a Biopharmaceutics Classification System class IV (BCS IV) drug known to possess a low oral bioavailability, poor permeability, and metabolism by cytochrome P450 (CYP) isozymes, such as CYP3A4 [53,54].

### 2.4. Molecular Docking

Among different mechanisms of action of drugs commonly used in BC hormone therapy, one of the most prevalent binds to and blocks estrogen receptors by estrogen receptors inhibitors. In this regard, the antiestrogen effect of UA has been recently described [55,56] as one of the mechanisms potentially responsible for the activity of UA against hormone-positive BC. To provide insight on the molecular level into the experimental differences in the activity of the tested compounds against both BC cell lines described in previous sections, the computational molecular docking of all newly synthesized compounds to the estrogen receptor was carried out.

The results of computational estimates of *K*_i_ values for the studied set of ligands and two models of estrogen receptor from two different crystal structures (PDB: 1QKG and 2YAT) revealed very strong binding to the estrogen receptor, similar to those of diosgenin and tigogenin conjugates previously studied by us [57]. To thoroughly analyze the results it is, however, necessary to compare the results with the poses of ligands found in both studied crystal structures. In both the case of estradiol (in the 1QKG structure) and estradiol-derived metal chelate (estradiolpyridine tetra acetate europium(III) in the 2YAT crystal structure), ligands were stabilized by two sets of hydrogen bonds. One of them is formed by E353 and R394, located deep in the binding pocket of the protein, with the hydroxyl group of the phenol part of estradiol, while the other is formed by H524, located close to the entrance of the binding pocket, with the hydroxyl group of cyclopentanol part of estradiol (see Figure 6). As all the new compounds studied in this work had a similar structure related to two functional groups capable of forming hydrogen bonds (the variable hydrophilic part on one end of the molecule and the acetate/carboxylate on the other end of the molecule), one may expect a similar binding. This is not always the case in the best poses shown in Table 4, as, depending on the length of the variable part of the ligands and their conformation, it was rarely possible to find a pose with both hydrogen bonds. Nevertheless, in the case of **12a** with the estimated most favorable Gibbs free energy of binding, we predicted the formation of two strong hydrogen bonds. First, between the amine moiety of the ligand and E353 and second between the carbonyl part of the ligand and R394 (see Figure 6). While in the pose, no hydrogen bond with H524 was identified due to the long distance to the ligand, this pose was stabilized by numerous van der Waals interactions in the binding pocket, resulting in a very low *K*_i_ value. It is also worth noting that, in some cases (e.g., **5a**), our molecular docking suggested an inverted pose, with the hydrophobic part close to the crucial E353/R394 residues and the hydrophilic part interacting with the backbones of residues located at the C-terminus of the receptor, such as K529. As the estimates of Gibbs free energy of binding for all studies compounds, regardless of the final pose, were similar, we could not determine the true pose of each of the studied ligands; although, poses with hydrogen bonds to Glu353/Arg394 seem to be more plausible.

In the case of compounds **4c** and **7a**, which showed the most promising results in the experimental part of this study, we predicted a similar pose to **12a** with hydrogen bonds to E353 and/or R394 (see Appendix A). However, for compound **10a** the predicted pose is more like the estradiol–pyridynium tetraacetic acid from the 2YAT crystal structure, but with no hydrogen bonds to E353/R394 residues. It is also worth noting that we found a strong affinity to estrogen receptors for most of the new ursolic acid derivatives, including those with high IC_50_ values against MCF-7 and MDA-MB-231 breast cancer cells, such as **3c**. As a result, we can suggest that the mechanism of biological action of some derivatives may be different resulting from direct interaction with estrogen receptors.

## 3. Materials and Methods

### 3.1. General Procedures

Ursolic acid was obtained from Wuhan Atomole Chemicals Co., Ltd. (Wuhan, China). Other materials, solvents, and reagents were of commercial origin and used without additional operations. Reactions were monitored on silica gel TLC plates 60 F_254_ (Merck, Darmstadt, Germany). Visualizations were performed with UV light (254 and/or 365 nm) then with CeMo stain and subsequent charring [58]. The melting points were determined using the Melting Point System (Mettler Toledo MP70, Metler Toledo, Greifensee, Switzerland). Solvents were evaporated under reduced pressure at 40 °C on the Büchi Rotavapor (BÜCHI Labortechnik AG, Flawil, Switzerland). Flash column chromatography was performed on silica gel (200–300 mesh).

The ^1^H NMR and ^13^C NMR spectra were acquired in CDCl_3_ (protected derivatives) and CD_3_OD (final structures) solutions on Bruker AVANCE III HD 500 MHz spectrometer (Bruker, Billerica, MA, USA) at the temperature 298 K. To identify the structures of all isolated products, analysis of the results of 1D and 2D NMR experiments was performed. The ^1^H and ^13^C NMR chemical shifts are given relative to the TMS signal at δ = 0.0 ppm.

Mass spectra were recorded on the MaldiSYNAPT G2S HDMS (Waters, Milford, MA, USA) spectrometer via electrospray ionization (ESI–MS). High-resolution mass spectrometry (HRMS) measurements were performed using the Synapt G2–Si mass spectrometer (Waters, Milford, MA, USA) equipped with an ESI source and a quadrupole-Time-of-flight mass analyzer. The results of the measurements were processed using the MassLynx 4.1 software (Waters, Milford, MA, USA).

### 3.2. Chemical Synthesis

#### 3.2.1. General Procedure for the Synthesis of Compounds **3**–**12**

##### Synthesis of 3β-Hydroxy-urs-12-en-28-oic Acid Methyl Ester (**2**)

K_2_CO_3_ (4.8 g, 35.0 mmol) and CH_3_I (2.4 mL, 38.7 mmol) were added to a solution of ursolic acid (8.0 g, 17.5 mmol) in DMF (80 mL). The mixture was stirred for 24 h at room temperature. After the reaction had been completed, DMF was evaporated under reduced pressure. Water (150 mL) was added to the residue and it was extracted with DCM (3 × 60 mL). The combined organic layers were dried over MgSO_4_, then filtered and evaporated to dryness. Product (**2**) was crystallized from hexane to give white crystals (7.4 g). Yield 90%; **Mp**: 173.2 °C; **^1^H NMR** δ [ppm]: **Ursolic (URS) part**: 0.71 (1H, m, H5), 0.74 (3H, s, H26), 0.78 (3H, s, H24), 0.85 (3H, d, H29), 0.92 (3H, s, H25), 0.93 (3H, d, H30), 0.98 (3H, s, H23), 0.98 (1H, m, H20), 0,98 (1H, m, H1), 1.06 (1H, m, H15), 1.07 (3H, s, H27), 1.29 (1H, m, H21), 1.31 (1H, m, H7), 1.33 (1H, m, H19), 1.35 (1H, m, H6), 1.48 (1H, m, H21), 1.49 (1H, m, H7), 1.49 (1H, m, H9), 1.52 (1H, m, H6), 1.58 (1H, m, H22), 1.58 (2H, m, H2), 1.63 (1H, m, H1), 1.66 (1H, m, H22), 1.67 (1H, m, H16), 1.76 (1H, m, H15), 1.90 (2H, m, H11), 1.99 (1H, m, H16), 2.22 (1H, d, H18), 3.21 (1H, m, H3), 3.60 (3H, s, OCH_3_), 5.24 (1H, m, H12); **^13^C NMR** δ [ppm]: **URS part**: 15.4 (C25), 15.6 (C24), 16.9 (C26), 17.0 (C29), 18.3 (C6), 21.2 (C30), 23.3 (C11), 23.6 (C27), 24.2 (C16), 27.2 (C2), 28.0 (C15), 28.1 (C23), 30.6 (C21), 33.0 (C7), 36.6 (C22), 37.0 (C10), 38.6 (C4), 38.7 (C1), 38.9 (C20), 39.0 (C19), 39.5 (C8), 42.0 (C14), 47.5 (C9), 48.1 (C17), 51.4 (OCH_3_), 52.9 (C18), 55.2 (C5), 79.0 (C3), 125.6 (C12), 138.1 (C13), 178.1 (C28); **HRMS**: [M+H]^+^ calcd for C_31_H_51_O_3_: 471.3838, found: 471.3829; [M+Na]^+^ calcd for C_31_H_50_O_3_Na: 493.3658, found: 493.3654.

##### Synthesis of Analogue **3**—3β-((*N*-(*t*-Butoxycarbonyl)-5-benzyl-*l*-glutamyl)oxy)-urs-12-en-28-oic Acid Methyl Ester

DMAP (39 mg, 0.319 mmol) and DCC (183 mg, 0.889 mmol) were added to the solution of Boc-*l*-Glu(Bzl)-OH (300 mg, 0.889 mmol) in anhydrous CH_2_Cl_2_ (15 mL) and the mixture was stirred for 10 min at room temperature. Then, ursolic acid methyl ester **2** (303 mg, 0.645 mmol) was added and the reaction mixture was stirred at room temperature for 24 h (TLC control). After the reaction had been completed, the DCU precipitate was filtered off. The filtrate was washed successively with NaHCO_3_ (5% aq.) and NaCl_aq_. The extract was dried over anhydrous MgSO_4_, filtered, and evaporated to dryness. The crude product was purified through a silica gel column eluted with ethyl acetate/hexane (9:1, *v*/*v*). Yield 60%, white foam; **^1^H NMR** δ [ppm]: ***l*-glutamic part**: 1.43 (9H, s, *t*-Bu), 1.95 (1H, m, H3′), 2.21 (1H, m, H3′), 2.43 (1H, m, H4′), 2.49 (1H, m, H4′), 4.33 (1H, m, H2′), 5.10 (1H, d, NH), 5.12 (2H, s, H6′), 7.38–7.29 (5H, m, Ph) + URS part; **^13^C NMR** δ [ppm]: ***l*-glutamic part**: 28.0 (C3′), 28.3 (OC(CH_3_)_3_), 30.4 (C4′), 53.2 (C2′), 66.4 (C6′), 79.9 (OC(CH_3_)_3_), {135.4 128.5 128.2 128.2} (Ph), 155.4 (NHCO), 171.6 (C1′), 172.9 (C5′) + URS part; **HRMS**: [M+H]^+^ calcd for C_36_H_58_NO_6_: 600.4264, found: 600.4267; [M–H]^+^ calcd for C_36_H_57_NO_6_Cl: 634.3874, found: 634.3887.

Compounds **4**–**9** were prepared according to this same synthetic protocol.

Compound **4**—3β-(*N*-(*t*-butoxycarbonyl)-*O*-benzyl-*l*-seryloxy)-urs-12-en-28-oic acid methyl ester. Yield 60%, white foam; **^1^H NMR** δ [ppm]: ***l*-serine part**: 1.44 (9H, s, *t*-Bu), 3.69 (1H, m, H3′), 3.92 (1H, m, H3′), 4.41 (1H, m, H2′), 4.49 (1H, m, H4′), 4.59 (1H, m, H4′), 5.42 (1H, d, NH), 7.35–7.25 (5H, m, Ph) + URS part; **^13^C NMR** δ [ppm]: ***l*-serine part**: 28.3 (OC(CH_3_)_3_), 54.2 (C2′), 70.5 (C3′), 73.3 (C4′), 79.8 (OC(CH_3_)_3_), {137.4, 128.3, 127.7, 127.7} (Ph), 155.5 (NHCO), 170.4 (C1′) + URS part; **HRMS**: [M+H]^+^ calcd for C_46_H_70_NO_7_: 748.5152, found: 748.5179; [M+Na]^+^ calcd for C_46_H_69_NO_7_Na: 770.4972, found: 770.4986.

Compound **5**—3β-(*N*^α^,*N*^τ^-bis-(*t*-butoxycarbonyl)-*l*-histidyloxy)-urs-12-en-28-oic acid methyl ester. Yield 46%, white foam; **^1^H NMR** δ [ppm]: ***l*-histidine part**: 1.42 (9H, s, *t*-Bu at OC2′(O)NH), 1.59 (9H, s, *t*-Bu at N7 of imidazole), 3.04 (2H, m, H3′), 4.54 (1H, m, H2′), 5.78 (1H, d, NH), 7.13 (1H, s, H8′), 7.98 (1H, s, H6′) + URS part; **^13^C NMR** δ [ppm]: ***l*-histidine part**: 27.9 (OC(CH_3_)_3_—imidazole), 28.4 (OC(CH_3_)_3_), 30.3 (C3′), 53.5 (C2′), 79.5 (OC(CH_3_)_3_), 85.5 (OC(CH_3_)_3_—imidazole), 114.6 (C8′—imidazole), 136.8 (C6′—imidazole), 138.8 (C4′—imidazole), 146.9 (NHCO—imidazole), 155.4 (NHCO), 171.1 (C1′) + URS part; **HRMS**: [M+H]^+^ calcd for C_47_H_74_N_3_O_8_: 808.5476, found: 808.5479; [M+Na]^+^ calcd for C_47_H_73_N_3_O_8_Na: 830.5295, found: 830.5292, [M+K]^+^ calcd for C_47_H_73_N_3_O_8_K: 846.5035, found: 846.5032.

Compound **6**—3β-(*N*-(*t*-butoxycarbonyl)-*l*-isoleucyloxy)-urs-12-en-28-oic acid methyl ester. Yield 70%, oil; **^1^H NMR** δ [ppm]: ***l*-isoleucine** part: 0.91 (3H, t, C6′), 0.93 (3H, d, C5′), 1.18 (1H, m, H4′), 1.44 (1H, m, H4′), 1.44 (9H, s, *t*-Bu), 1.89 (1H, m, H3′), 4.26 (1H, dd, H2′), 5.00 (1H, d, NH) + URS part; **^13^C NMR** δ [ppm]: ***l*-isoleucine** part: 11.7 (C5′), 15.5 (C6′), 24.7 (C4′), 28.3 (OC(CH_3_)_3_), 38.0 (C3′), 58.4 (C2′), 79.6 (OC(CH_3_)_3_), 155.6 (NHCO), 172.0 (C1′) + URS part; **HRMS**: [M+H]^+^ calcd for C_42_H_70_NO_6_: 684.5203, found: 684.5216; [M+Na]^+^ calcd for C_42_H_69_NO_6_Na: 706.5023, found: 706.5035.

Compound **7**—3β-(*N*-(*t*-butoxycarbonyl)-*l*-prolyloxy)-urs-12-en-28-oic acid methyl ester. Yield 81%, white foam; **^1^H NMR** δ [ppm], two conformers are observed: (**1**) ***l*-proline part**: 1.41 (9H, s, *t*-Bu), 1.85 and 1.93 (2H, 2×m, both H4′), 1.96 and 2.23 (2H, 2×m, both H3′), 3.45 and 3.53 (2H, 2×m, both H5′), 4.23 (1H, m, H2′) + URS part; (**2**) 1.44 (9H, s, *t*-Bu), 1.87 and 1.94 (2H, 2×m, both H4′), 1.94 and 2.18 (2H, 2×m, both H3′), 3.39 and 3.48 (2H, 2×m, both H5′), 4.31 (1H, m, H2′) + URS part; **^13^C NMR** δ [ppm], two conformers are observed: (**1**) ***l*-proline part**: 23.5 (C4′), 28.4 (OC(CH_3_)_3_), 31.1 (C3′), 46.2 (C5′), 59.4 (C2′), 79.9 (OC(CH_3_)_3_), 154.9 (NHCO), 172.8 (C1′) + URS part; (**2**) 24.3 (C4′), 28.5 (OC(CH_3_)_3_), 30.1 (C3′), 46.4 (C5′), 59.3 (C2′), 79.5 (OC(CH_3_)_3_), 154.2 (NHCO), 172.7 (C1′) + URS part. **HRMS**: [M+H]^+^ calcd for C_41_H_66_NO_6_: 668.4890, found: 668.4901; [M+Na]^+^ calcd for C_41_H_65_NO_6_Na: 690.4710, found: 690.4715.

Compound **8**—3β-(*N*-(*t*-butoxycarbonyl)-*l*-leucyloxy)-urs-12-en-28-oic acid methyl ester. Yield 51%, white foam; **^1^H NMR** δ [ppm]: ***l***-**leucine part**: 0.94 and 0.95 (6H, 2×d, both H5′), 1.44 (9H, s, *t*-Bu), 1.48 (1H, m, H3′), 1.60 (1H, m, H3′), 1.73 (1H, m, H4′), 4.28 (1H, m, H2′), 4.88 (1H, d, NH); **Ursolic (URS) part**: 0.74 (3H, s, H26), 0.82 (1H, m, H5), 0.86 (3H, s, H23), 0.86 (3H, s, H24), 0.86 (3H, d, H29), 0.94 (3H, s, H25), 0.94 (3H, d, H30), 1.00 (1H, m, H20), 1.06 (1H, m, H15), 1,07 (1H, m, H1), 1.07 (3H, s, H27), 1.30 (1H, m, H21), 1.31 (1H, m, H7), 1.32 (1H, m, H19), 1.36 (1H, m, H6), 1.48 (1H, m, H21), 1.49 (1H, m, H7), 1.50 (1H, m, H6), 1.52 (1H, m, H9), 1.59 (1H, m, H22), 1.65 (2H, m, H2), 1.66 (1H, m, H22), 1.67 (1H, m, H1), 1.67 (1H, m, H16), 1.77 (1H, m, H15), 1.91 (2H, m, H11), 2.00 (1H, m, H16), 2.24 (1H, d, H18), 3.60 (3H, s, H31), 4.52 (1H, m, H3), 5.25 (1H, m, H12); **^13^C NMR** δ [ppm]: ***l*-leucine** part: 21.9 and 23.0 (both C5′), 24.8 (C4′), 28.3 (OC(CH_3_)_3_), 42.1 (C3′), 52.5 (C2′), 79.6 (OC(CH_3_)_3_), 155.4 (NHCO), 173.2 (C1′); **Ursolic (URS) part**: 15.4 (C25), 16.8 (C24), 16.9 (C26), 17.0 (C29), 18.1 (C6), 21.2 (C30), 23.3 (C11), 23.6 (C27), 23.5 (C2), 24.2 (C16), 28.0 (C15), 28.2 (C23), 30.6 (C21), 32.9 (C7), 36.6 (C22), 36.8 (C10), 37.8 (C4), 38.2 (C1), 38.9 (C20), 39.0 (C19), 39.5 (C8), 42.0 (C14), 47.4 (C9), 48.1 (C17), 51.4 (C31), 52.8 (C18), 55.3 (C5), 82.0 (C3), 125.4 (C12), 138.2 (C13), 178.0 (C28); **HRMS**: [M+H]^+^ calcd for C_42_H_70_NO_6_: 684.5203, found: 684.5219; [M+Na]^+^ calcd for C_42_H_69_NO_6_Na: 706.5023, found: 706.5033.

Compound **9**—3β-(*N*-(*t*-butoxycarbonyl)-*l*-phenylalanyloxy)-urs-12-en-28-oic acid methyl ester. Yield 70%, white foam; **^1^H NMR** δ [ppm]: ***l*-phenylalanine part**: 1.39 (9H, s, *t*-Bu), 3.02 (1H, m, H3′), 3.13 (1H, m, H3′), 4.51 (1H, m, H2′), 4.89 (1H, d, NH), 7.17, 7.22, 7.28 (5H, m, Ph) + URS part; **^13^C NMR** δ [ppm]: ***l*-phenylalanine part**: 28.3 (OC(CH_3_)_3_), 38.5 (C3′), 54.6 (C2′), 79.7 (OC(CH_3_)_3_), {136.1, 129.4, 128.5, 126.9} (Ph), 155.0 (NHCO), 171.8 (C1′) + URS part; **HRMS**: [M+H]^+^ calcd for C_45_H_68_NO_6_: 718.5047, found: 718.5063.

##### Synthesis of Analogue **10**—3β-(*N*-(*t*-Butoxycarbonyl)-*l*-alanyl-*l*-isoleucyloxy)-urs-12-en-28-oic Acid Methyl Ester

DIPEA (83 µL, 0.474 mmol) was added to the solution of Boc-*l-*Ala-OH (30 mg, 0.157 mmol), TBTU (101 mg, 0.314 mmol) and HOBt (42 mg, 0.314 mmol) in DCM (5 mL), and the mixture was stirred for 15 min at room temperature. Then, Ile-Urs-OMe **14** (110 mg, 0.188 mmol) was added, and the reaction mixture was stirred at room temperature for 24 h (TLC monitoring). After the reaction had been completed, DCM (15 mL) was added. The organic layer was washed successively with 5% NaHCO_3 aq._ and then NaCl _aq_. The extract was dried over anhydrous MgSO_4_, filtered, and evaporated to dryness. The crude product was purified through a silica gel column eluted with ethyl acetate/hexane (9:1, *v*/*v*). Yield 80%, oil; **^1^H NMR** δ [ppm]: ***l*-alanine-*l*-isoleucine** part: 0.91 (3H, t, C6′), 0.93 (3H, d, C5′), 1.17 (1H, m, H4′), 1.35 (3H, d, H9′), 1.44 (1H, m, H4′), 1.44 (9H, s, *t*-Bu), 1.85 (1H, m, H3′), 4.17 (1H, m, H8′), 4.57 (1H, m, H2′), 4.96 (1H, d, NHOC(CH_3_)_3_), 6.62 (1H, d, NH) + URS part; **^13^C NMR** δ [ppm]: ***l*-alanine-*l*-isoleucine** part: 11.6 (C5′), 15.6 (C6′), 17.7 (C9′), 24.6 (C4′), 28.3 (OC(CH_3_)_3_), 37.9 (C3′), 50.0 (C8′), 57.9 (C2′), 80.1 (OC(CH_3_)_3_), 155.6 (NHCO), 171.3 (C1′), 172.3 (C7′) + URS part; **HRMS**: [M+H]^+^ calcd for C_45_H_75_N_2_O_7_: 755.5574, found: 755.5571.

Compounds **11** and **12** were prepared according to this same synthetic protocol.

Compound **11**—3β-(*N*-(*t*-butoxycarbonyl)-*l*-isoleucyl-*l*-prolyloxy)-urs-12-en-28-oic acid methyl ester. Yield 87%, oil; **^1^H NMR** δ [ppm]: ***l*-isoleucine-*l*-proline** part: 0.90 (3H, t, H10′), 1.14 (1H, m, H8′), 1.14 (3H, d, H11′), 1.41 (9H, s, *t*-Bu), 1.60 (1H, m, H9′), 1.76 (1H, m, H8′), 1.96 (1H, m, H3′), 1.97 (1H, m, H4′), 2.04 (1H, m, H4′), 2.22 (1H, m, H3′), 3.65 (1H, m, H5′), 3.81 (1H, m, H5′), 4.28 (1H, m, H7′), 4.50 (1H, m, H2′), 5.15 (1H, d, NH’) + URS part; **^13^C NMR** δ [ppm]: ***l*-isoleucine-*l*-proline** part: 11.2 (C10′), 15.4 (C11′), 24.2 (C9′), 25.0 (C4′), 28.3 (OC(CH_3_)_3_), 29.2 (C3′), 37.9 (C8′), 47.2 (C5′), 56.3 (C7′), 59.2 (C2′), 79.4 (OC(CH_3_)_3_), 155.8 (NHCO), 171.4 (C6′), 171.7 (C1′) + URS part; **HRMS**: [M+H]^+^ calcd for C_47_H_77_N_2_O_7_: 781.5731, found: 781.5741.

Compound **12**—3β-(*N*-(*t*-butoxycarbonyl)-*l*-leucyl-*l*-prolyloxy)-urs-12-en-28-oic acid methyl ester. Yield 81%, oil; **^1^H NMR** δ [ppm]: ***l*-leucine-*l*-proline** part: 0.94 and 0.93 (6H, 2×d, both H10′), 1.44 (9H, s, *t*-Bu), 1.48 (1H, m, H8′), 1.60 (1H, m, H8′), 1.73 (1H, m, H9′), 1.97 (1H, m, H4′), 1.97 (1H, m, H3′), 2.04 (1H, m, H4′), 2.22 (1H, m, H3′), 3.59 (1H, m, H5′), 3.75 (1H, m, H5′), 4.47 (1H, m, H7′), 4.50 (1H, m, H2′), 5.15 (1H, d, NH) + URS part; **^13^C NMR** δ [ppm]: ***l*-leucine-*l*-proline** part: 23.4 and 21.6 (both C10′), 24.6 (C7′), 24.9 (C4′), 28.3 (OC(CH_3_)_3_), 29.2 (C3′), 42.2 (C8′), 46.7 (C5′), 50.3 (C7′), 59.2 (C2′), 79.4 (OC(CH_3_)_3_), 155.8 (NHCO), 171.7 (C6′), 171.7 (C1′) + URS part; **HRMS**: [M+H]^+^ calcd for C_47_H_77_N_2_O_7_: 781.5731, found: 781.5745.

#### 3.2.2. General Procedure for the Synthesis of Compounds **5a**–**12a**

##### Synthesis of **5a**—3β-(*l*-Histidyloxy)-urs-12-en-28-oic Acid Methyl Ester Hydrochloride

Compound **5** (209 mg, 0.259 mmol) was dissolved in AcOEt (3 mL) and treated with 2.9 M HCl_g_/AcOEt (5.8 mM, 3 mL) and stirred for 24 h (TLC monitoring). The precipitated salt **5a** was collected by filtration and dried under vacuum at room temperature. Yield 82%, white solid; **Mp**: 220.7 °C (dec.); **^1^H NMR** δ [ppm]: ***l*-histidine part**: 3.41 (1H, m, H3′), 4.52 (1H, t, H2′), 7.59 (1H, s, H8′), 8.98 (1H, d, H6′); **URS part**: 0.79 (3H, s, H26), 0.89 (1H, m, H5), 0.91 (3H, d, H29), 0.96 (3H, s, H23), 0.96 (3H, s, H24), 0.99 (3H, s, H30), 0.99 (1H, m, H20), 1.02 (3H, s, H25), 1.12 (1H, m, H1), 1.11 (1H, m, H15), 1.15 (3H, s, H27), 1.37 (1H, m, H21), 1.37 (2H, m, H7), 1.40 (1H, m, H19), 1.45 (1H, m, H6), 1.52 (1H, m, H21), 1.57 (1H, m, H7), 1.57 (1H, m, H6), 1.62 (1H, m, H22), 1.62 (1H, m, H9), 1.68 (1H, m, H22), 1.68 (1H, m, H16), 1.74 (1H, m, H1), 1.77 (2H, m, H2), 1.83 (1H, m, H15), 1.96 (2H, m, H11), 2.08 (1H, m, H16), 2.25 (1H, d, H18), 3.61 (3H, s, H31), 4.69 (1H, m, H3), 5,26 (1H, t, H12).; **^13^C NMR** δ [ppm]: ***l*-histidine part**: 28.88 (C3′), 53.00 (C2′), 119.68 (C8′), 128.50 (C4′), 136.11 (C6′), 168.99 (C1′); **URS part**: 15.94 (C25), 17.23 (C24), 17.61 (C26), 17.61 (C29), 19.21 (C6), 21.49 (C30), 24.14 (C27), 24.31 (C11), 24.43 (C2), 25.25 (C16), 28.62 (C23), 29.11 (C15), 31.62 (C21), 34.02 (C7), 37.85 (C22), 37.98 (C10), 38.77 (C4), 39.29 (C1), 40.31 (C19), 40.34 (C20), 40.76 (C8), 43.16 (C14), 48.76 (C9), 49.50 (C17), 52.09 (C31), 54.35 (C18), 56.54 (C5), 86.06 (C3), 126.81 (C12), 139.60 (C13), 179.74 (C28); **HRMS** (ESI) m/z: [M–H]^–^ calcd for C_37_H_57_N_3_O_4_Cl: 642.4038, found: 642.4033; [M+H]^+^ calcd for C_37_H_58_N_3_O_4_: 608.4427, found: 608.4433.

Compound **6a**—3β-(*l*-isoleucyloxy)-urs-12-en-28-oic acid methyl ester hydrochloride. Yield 60%, white solid; **Mp**: 250.2 °C (dec.); **^1^H NMR**: δ [ppm]: ***l*-isoleucine part**: 1.03 (3H, t, H5′), 1.10 (3H, d, H6′), 1.42 (1H, m, H4′), 1.57 (1H, m, H4′), 2.05 (1H, m, H3′), 4.04 (1H, d, H2′); **URS part**: 0.82 (3H, s, H26), 0.91 (3H, d, H29), 0.93 (1H, m, H5), 0.97 (3H, s, H23), 0.96 (3H, s, H24), 0.99 (3H, d, H30), 0.99 (1H, m, H20), 1.03 (3H, s, H25), 1.12 (1H, m, H1), 1.12 (1H, m, H15), 1.16 (3H, s, H27), 1.37 (1H, m, H21), 1.38 (2H, m, H7), 1.41 (1H, m, H19), 1.48 (1H, m, H6), 1.48 (1H, m, H7), 1.52 (1H, m, H21), 1.60 (1H, m, H6), 1.60 (1H, m, H22), 1.63 (1H, m, H9), 1.67 (1H, m, H22), 1.68 (1H, m, H16), 1.74 (1H, m, H1), 1.76 (2H, m, H2), 1.83 (1H, m, H15), 1.94 (2H, m, H11), 2.09 (1H, m, H16), 2.25 (1H, d, H18), 3.62 (3H, s, H31), 4.67 (1H, m, H3), 5,26 (1H, t, H12); **^13^C NMR**: δ [ppm]: ***l*-isoleucine part**: 12.04 (C5′), 15.19 (C6′), 26.40 (C4′), 37.79 (C3′), 58.89 (C2′), 168.32 (C1′); **URS part**: 15.94 (C25), 17.47 (C24), 17.63 (C26), 17.63 (C29), 19.27 (C6), 21.60 (C30), 24.17 (C27), 24.34 (C11), 24.56 (C2), 25.27 (C16), 28.89 (C23), 29.12 (C15), 31.63 (C21), 34.04 (C7), 37.86 (C22), 37.99 (C10), 38.72 (C4), 39.33 (C1), 40.32 (C19), 40.35 (C20), 43.18 (C8), 43.18 (C14), 48.78 (C9), 48.45 (C17), 52.10 (C31), 54.36 (C18), 56.62 (C5), 86.55 (C3), 126.84 (C12), 139.61 (C13), 179.76 (C28); **HRMS** (ESI) m/z: [M+H]^+^ calcd for C_37_H_62_NO_4_: 584.4679, found: 584.4691.

Compound **7a**—3β-(*l*-prolyloxy)-urs-12-en-28-oic acid methyl ester hydrochloride. Yield 82%, white solid; **Mp**: 260.1 °C (dec.); **^1^H NMR** δ [ppm]: ***l*-proline part**: 2.03 (1H, m, H4′), 2.14 (1H, m, H4′), 2.17 (1H, m, H3′), 2.46 (1H, m, H3′), 3.55 (2H, m, H5′), 4.46 (1H, m, H2′); **URS part**: 0.73 (3H, s, H26), 0.80 (1H, m, H5), 0.86 (3H, d, H28), 0.86 (3H, s, H23), 0.86 (3H, s, H24), 0.94 (3H, d, H29), 0.94 (3H, s, H25), 1.00 (1H, m, H20), 1.02 (1H, m, H1), 1.06 (1H, m, H15), 1.07 (3H, s, H27), 1.29 (1H, m, H21), 1.32 (2H, m, H7), 1.32 (1H, m, H19), 1.36 (1H, m, H6), 1.47 (2H, m, H7), 1.48 (1H, m, H21), 1.49 (1H, m, H6), 1.50 (1H, m, H9), 1.59 (1H, H22), 1.67 (1H, m, H1), 1.66 (1H, H22), 1.66 (1H, m, H16), 1.69 (2H, m, H2), 1.77 (1H, m, H15), 1.89 (2H, m, H11), 2.00 (1H, m, H16), 2.23 (1H, d, H18), 3.60 (3H, s, H31), 4.61 (1H, m, H3), 5,23 (1H, t, H12); **^13^C NMR** δ [ppm]: ***l*-proline** part: 23.7 (C4′), 29.1 (C3′), 45.8 (C5′), 59.3 (C2′), 168.7 (C1′); **URS part**: 15.4 (C25), 16.8 (C24), 16.9 (C26), 17.1 (C28), 18.1 (C6), 21.2 (C29), 23.3 (C11), 23.5 (C2), 23.6 (C27), 24.2 (C16), 28.0 (C15), 28.3 (C23), 30.6 (C21), 32.8 (C7), 36.6 (C22), 36.8 (C10), 37.9 (C4), 38.2 (C1), 38.8 (C20), 39.0 (C19), 39.5 (C8), 42.0 (C14), 47.4 (C9), 48.1 (C17), 51.4 (C31), 52.8 (C18), 55.2 (C5), 84.4 (C3), 125.3 (C12), 138.2 (C13), 178.0 (C30); **HRMS** (ESI) m/z: [M+H]^+^ calcd for C_36_H_58_NO_4_: 568.4366, found: 568.4374.

Compound **8a**—3β-(*l*-leucyloxy)-urs-12-en-28-oic acid methyl ester hydrochloride. Yield 95%, white solid; **Mp**: 261.2 °C (dec.); **^1^H NMR** δ [ppm]: ***l*-leucine part**: 1.02 (3H, d, H5′), 1.04 (3H, d, H5′), 1.69 (1H, m, H3′), 1.83 (1H, m, H3′), 1.84 (1H, m, H4′), 4.03 (1H, m, H2′); **URS part**: 0.80 (3H, s, H26), 0.91 (3H, d, H29), 0.93 (1H, m, H5), 0.95 (3H, s, H23), 0.95 (3H, s, H24), 0.99 (3H, m, H30), 0.99 (1H, m, H20), 1.03 (3H, s, H25), 1.12 (1H, m, H1), 1.12 (1H, m, H15), 1.15 (3H, s, H27), 1.37 (1H, m, H21), 1.38 (2H, m, H7), 1.40 (1H, m, H19), 1.46 (1H, m, H6), 1.52 (1H, m, H21), 1.57 (1H, m, H6), 1.58 (1H, m, H7), 1.60 (1H, m, H22), 1.61 (1H, m, H9), 1.67 (1H, m, H22), 1.69 (1H, m, H16), 1.75 (1H, m, H1), 1.74 (2H, m, H2), 1.82 (1H, m, H15), 1.96 (2H, m, H11), 2.08 (1H, m, H16), 2.25 (1H, d, H18), 3.62 (3H, s, H31), 4.67 (1H, m, H3), 5.26 (1H, t, H12); **^13^C NMR** δ [ppm]: ***l*-leucine part**: 22.20 (C5′), 22.65 (C5′), 25.83 (C4′), 41.05 (C3′), 52.75 (C2′), 171.06 (C1′); **URS part**: 15.96 (C25), 17.35 (C24), 17.63 (C26), 17.63 (C29), 19.25 (C6), 21.60 (C30), 24.16 (C27), 24.34 (C11), 24.49 (C2), 25.27 (C16), 28.77 (C23), 29.12 (C15), 31.63 (C21), 34.03 (C7), 37.86 (C22), 38.01 (C10), 38.87 (C4), 39.33 (C1), 40.32 (C19), 40.35 (C20), 40.78 (C8), 43.18 (C14), 48.80 (C9), 49.45 (C17), 52.10 (C31), 54.36 (C18), 56.61 (C5), 85.19 (C3), 126.84 (C12), 139.60 (C13), 179.75 (C28); **HRMS** (ESI) m/z: [M+H]^+^ calcd for C_37_H_62_NO_4_: 584.4679, found: 584.4686.

Compound **9a**—3β-(*l*-phenylalanyloxy)-urs-12-en-28-oic acid methyl ester hydrochloride. Yield 96%, white solid; **Mp**: 263.8 °C (dec.); **^1^H NMR** δ [ppm]: ***l*-phenylalanine part**: 3.14 (1H, m, H3′), 3.31 (1H, m, H3′), 4.36 (1H, t, H2′), 7.32 (2H, s, H5′), 7.39 (2H, s, H6′), 7.33 (1H, m, H7′); **URS part**: 10.76 (3H, s, H23), 0.76 (3H, s, H26), 0.79 (3H, s, H24), 0.87 (1H, m, H5), 0.87 (3H, d, H29), 0.98 (3H, s, d, H30), 0.99 (1H, m, H20), 1.00 (3H, s, H25), 1.09 (1H, m, H1), 1.10 (1H, m, H15), 1.14 (3H, s, H27), 1.35 (1H, m, H21), 1.35 (2H, m, H7), 1.40 (1H, m, H19), 1.42 (1H, m, H6), 1.50 (1H, m, H21), 1.56 (1H, m, H7), 1.54 (1H, m, H6), 1.60 (1H, m, H22), 1.61 (1H, m, H9), 1.66 (1H, m, H22), 1.68 (1H, m, H16), 1.73 (1H, m, H1), 1.73 (2H, m, H2), 1.81 (1H, m, H15), 1.96 (2H, m, H11), 2.06 (1H, m, H16), 2.25 (1H, d, H18), 3.61 (3H, s, H31), 4.61 (1H, m, H3), 5,25 (1H, m, H12); **^13^C NMR** δ [ppm]: ***l*-phenylalanine part**: 37.74 (C3′), 55.32 (C2′), 129.01 (C7′), 130.28 (C6′), 130.35 (C5′), 135.33 (C4′), 170.14 (C1′); **URS part**: 15.94 (C25), 17.12 (C24), 17.62 (C26), 17.62 (C29), 19.21 (C6), 21.50 (C30), 24.15 (C27), 24.32 (C11), 24.45 (C2), 25.26 (C16), 28.67 (C23), 29.12 (C15), 31.63 (C21), 34.02 (C7), 37.86 (C22), 37.97 (C10), 38.74 (C4), 39.32 (C1), 40.32 (C19), 40.35 (C20), 40.77 (C8), 43.17 (C14), 48.79 (C9), 49.50 (C17), 52.09 (C31), 54.36 (C18), 56.62 (C5), 85.81 (C3), 126.84 (C12), 139.60 (C13), 179.75 (C28); **HRMS** (ESI) m/z: [M+H]^+^ calcd for C_40_H_60_NO_4_: 618.4522, found: 618.4530.

Compound **10a**—3β-(*l*-alanyl-*l*-isoleucyloxy)-urs-12-en-28-oic acid methyl ester hydrochloride. Yield 95%, white solid; **Mp**: 186.4 °C (dec.); **^1^H NMR** δ [ppm]: ***l*-alanine-*l*-isoleucine part**: 0.97 (3H, t, H5′), 1.01 (3H, d, H6′), 1.30 (1H, m, H4′), 1.61 (1H, m, H4′), 1.65 (3H, d, H9′), 1.98 (1H, m, H3′), 4.04 (1H, q, H8′), 4.46 (1H, d, H2′); **URS part**: 0.80 (3H, s, H26), 0.89 (1H, m, H5), 0.90 (3H, d, H29), 0.92 (3H, s, H23), 0.94 (3H, s, H24), 0.97 (3H, s, H30), 1.00 (1H, m, H20), 1.01 (3H, s, H25), 1.08 (1H, m, H1), 1.10 (1H, m, H15), 1.15 (3H, s, H27), 1.36 (2H, m, H7), 1.36 (1H, m, H21), 1.40 (1H, m, H19), 1.44 (1H, m, H6), 1.51 (1H, m, H21), 1.57 (1H, m, H6), 1.57 (1H, m, H7), 1.60 (1H, m, H9), 1.58 (1H, m, H22), 1.66 (2H, m, H2), 1.65 (1H, m, H22), 1.68 (1H, m, H16), 1.71 (1H, m, H1), 1.81 (1H, m, H15), 1.95 (2H, m, H11), 2.08 (1H, m, H16), 2.25 (1H, d, H18), 3.62 (3H, s, H31), 4.56 (1H, m, H3), 5.26 (1H, t, H12); **^13^C NMR** δ [ppm]: ***l*-alanine-*l*-isoleucine part**: 11.68 (C5′), 16.15 (C6′), 26.06 (C4′), 38.24 (C3′), 50.04 (C8′), 59.07 (C2′), 172.33 (C1′), 171.20 (C7′); **URS part**: 15.98 (C25), 17.48 (C24), 17.63 (C26), 17.63 (C29), 17.94 (C6′), 19.27 (C6), 21.50 (C30), 24.16 (C27), 24.34 (C11), 24.63 (C2), 25.22 (C16), 28.82 (C23), 29.12 (C15), 31.63 (C21), 34.06 (C7), 37.86 (C22), 38.02 (C10), 38.78 (C4), 39.39 (C1), 40.32 (C19), 40.35 (C20), 40.79 (C8), 43.17 (C14), 48.81 (C9), 49.50 (C17), 52.09 (C31), 54.36 (C18), 56.71 (C5), 83.72 (C3), 126.88 (C12), 139.58 (C13), 179.76 (C28); **HRMS** (ESI) m/z: [M+H]^+^ calcd for C_40_H_67_N_2_O_5_: 655.5050, found: 655.5053.

Compound **11a**—3β-(*l*-isoleucyl-*l*-prolyloxy)-urs-12-en-28-oic acid methyl ester hydrochloride. Yield 53%, white solid; **Mp**: 207.8 °C (dec.); **^1^H NMR** δ [ppm]: ***l*-isoleucine-*l*-proline part**: 1.01 (1H, m, H9′), 1.02 (3H, t, H10′), 1.16 (3H, d, H11′), 1.26 (1H, m, H9′), 2.02 (1H, m, H8′), 2.03 (1H, m, H3′), 2.06 (2H, m, H4′), 2.36 (1H, m, H3′), 3.66 (1H, m, H5′), 3.78 (1H, m, H5′), 4.12 (1H, m, H7′), 4.56 (1H, m, H2′); **URS part**: 0.80 (3H, s, H26), 0.80 (1H, m, H5), 0.91 (3H, d, H29), 0.90 (3H, s, H23), 0.93 (3H, s, H24), 0.99 (3H, s, H30), 1.01 (1H, m, H20), 1.02 (3H, s, H25), 1.09 (1H, m, H1), 1.11 (1H, m, H15), 1.15 (3H, s, H27), 1.37 (1H, m, H21), 1.37 (2H, m, H7), 1.41 (1H, m, H19), 1.49 (1H, m, H6), 1.52 (1H, m, H21), 1.57 (1H, m, H7), 1.61 (1H, m, H9), 1.60 (1H, m, H22), 1.68 (1H, m, H22), 1.69 (1H, m, H16), 1.69 (1H, m, H6), 1.70 (2H, m, H2), 1.72 (1H, m, H1), 1.83 (1H, m, H15), 1.97 (2H, m, H11), 2.09 (1H, m, H16), 2.25 (1H, d, H18), 3.62 (3H, s, H31), 4.66 (1H, m, H3), 5,26 (1H, t, Hz, H12); **^13^C NMR** δ [ppm]: ***l*-isoleucine-*l*-proline part**: 11.66 (C10′), 15.19 (C11′), 24.72 (C9′), 26.12 (C4′), 37.69 (C8′), 57.47 (C7′), 60.98 (C2′), 168.70 (C6′), 172.80 (C1′); **URS part**: 16.02 (C25), 17.32 (C24), 17.64 (C29), 17.64 (C26), 19.26 (C6), 21.51 (C30), 24.18 (C27), 24.34 (C11), 24.56 (C2), 25.28 (C16), 28.72 (C23), 30.23 (C3′), 29.13 (C15), 31.63 (C21), 34.06 (C7), 37.86 (C22), 38.04 (C10), 38.95 (C4), 39.41 (C1), 40.33 (C19), 40.31 (C20), 40.79 (C8), 43.17 (C14), 46.77 (C5′), 48.82 (C9), 49.50 (C17), 52.09 (C31), 54.37 (C18), 55.66 (C5), 83.38 (C3), 126.89 (C12), 139.58 (C13), 179.76 (C28); **HRMS** (ESI) m/z: [M+H]^+^ calcd for C_42_H_69_N_2_O_5_: 681.5206, found: 681.5219.

Compound **12a**—3β-(*l*-leucyl-*l*-prolyloxy)-urs-12-en-28-oic acid methyl ester hydrochloride. Yield 93%, white solid; **Mp**: 185.3 °C (dec.); **^1^H NMR** δ [ppm]: ***l*-leucine-*l*-proline part**: 1.07 (6H, d, H10′), 1.72 (1H, m, H8′), 1.83 (1H, m, H9′), 2.01 (1H, m, H3′), 2.08 (1H, m, H4′), 2.34 (1H, m, H3′), 3.57 (1H, m, H5′), 3.76 (1H, m, H5′), 4.23 (1H, m, H7′), 4.54 (1H, m, H2′); **URS part**: 0.80 (3H, s, H24), 0.80 (3H, s, H26), 0.88 (1H, m, H5), 0.90 (3H, d, H29), 0.90 (3H, s, H23), 0.97 (3H, s, H30), 1.00 (1H, m, H20), 1.02 (3H, s, H25), 1.07 (1H, m, H1), 1.11 (1H, m, H15), 1.15 (3H, s, H27), 1.36 (2H, m, H7), 1.36 (1H, m, H21), 1.39 (1H, m, H19), 1.44 (1H, m, H6), 1.51 (1H, m, H21), 1.57 (1H, m, H7), 1.57 (1H, m, H6), 1.60 (1H, m, H9), 1.60 (1H, m, H22), 1.67 (2H, m, H2), 1.67 (1H, m, H22), 1.67 (1H, m, H16), 1.71 (1H, m, H1), 1.81 (1H, m, H15), 1.94 (2H, m, H11), 2.07 (1H, m, H16), 2.25 (1H, d, H18), 3.62 (3H, s, H31), 4.54 (1H, m, H3), 5,26 (1H, t, H12); **^13^C NMR** δ [ppm]: ***l*-leucine-*l*-proline part**: 21.47 (C10′), 25.38 (C9′), 26.04 (C4′), 28.45 (C5′), 30.22 (C3′), 41.03 (C8′), 51.53 (C7′), 61.06 (C2′), 169.36 (C6′), 172.88 (C1′); **URS part**: 16.00 (C25), 17.39 (C29), 17.62 (C24), 17.62 (C26), 19.24 (C6), 21.60 (C30), 24.15 (C27), 24.33 (C11), 24.48 (C2), 25.27 (C16), 28.73 (C23), 29.12 (C15), 31.63 (C21), 34.05 (C7), 37.86 (C22), 38.03 (C10), 38.90 (C4), 39.38 (C1), 40.32 (C19), 40.35 (C20), 40.78 (C8), 43.17 (C14), 48.81 (C9), 49.50 (C17), 52.09 (C31), 54.36 (C18), 56.70 (C5), 83.43 (C3), 126.88 (C12), 139.58 (C13), 179.77 (C28); **HRMS** (ESI) m/z: [M+H]^+^ calcd for C_42_H_69_N_2_O_5_: 681.5206, found: 681.5217.

#### 3.2.3. Synthesis of Compounds **3b** and **4b**

Compound **3b**—3β-(*N*-(*t*-Butoxycarbonyl)-*l*-glutamyloxy)-urs-12-en-28-oic acid methyl ester.

A solution of **3** (345 mg, 0.437 mmol) in 25 mL AcOEt was hydrogenated in the presence of 10% palladium on charcoal (70 mg) for 4 h. After filtration through a cellite pad, the solution was evaporated. The product was obtained (295 mg) as colorless oil. Yield 96%. The crude product was used in the next step without further purification.

Compound **4b**—3β-(*N*-(*t*-butoxycarbonyl)-*l*-seryloxy)-urs-12-en-28-oic acid methyl ester.

A solution of **4** (90 mg, 0.114 mmol) in 20 mL AcOEt was hydrogenated in the presence of 10% palladium on charcoal (40 mg) for 6 h. After filtration through a cellite pad, the solution was evaporated. The product was obtained (72 mg) as colorless oil. Yield 96%. The crude product was used in the next step without further purification.

#### 3.2.4. Synthesis of Compounds **3c** and **4c**

Compound **3c**—3β-(*l*-Glutamyloxy)-urs-12-en-28-oic acid methyl ester hydrochloride.

Compound **3b** (250 mg, 0.357 mmol) was dissolved in AcOEt (3 mL), treated with 2.9 M HCl_g_/AcOEt (3 mL), and stirred for 24 h (TLC monitoring). The precipitated salt was collected by filtration and dried under vacuum at room temperature. Yield 65%, white solid; **Mp**: 219.8 °C (dec.); **^1^H NMR** δ [ppm]: ***l*-glutamic part**: 2.15 (1H, m, H3′), 2.28 (1H, m, H3′), 2.58 (2H, m, H4′), 4.16 (1H, t, H2′); **URS part**: 0.81 (3H, s, H26), 0.91 (3H, d, H29), 0.92 (1H, m, H5), 0.96 (3H, s, H23), 0.97 (3H, s, H24), 0.98 (3H, s, H30), 0.99 (1H, m, H20), 1.03 (3H, s, H25), 1.11 (1H, m, H1), 1.10 (1H, m, H15), 1.16 (3H, s, H27), 1.36 (1H, m, H21), 1.40 (1H, m, H19), 1.49 (1H, m, H6), 1.49 (2H, m, H7), 1.51 (1H, m, H21), 1.58 (1H, m, H7), 1.61 (1H, m, H22), 1.62 (1H, m, H9), 1.67 (1H, m, H22), 1.69 (1H, m, H6), 1.69 (1H, m, H16), 1.74 (1H, m, H1), 1.75 (2H, m, H2), 1.82 (1H, m, H15), 1.96 (2H, m, H11), 2.08 (1H, m, H16), 2.25 (1H, m, H18), 3.62 (3H, s, H31), 4.69 (1H, m, H3), 5.26 (1H, m, H12); **^13^C NMR** δ [ppm]: ***l*-glutamic part**: 30.40 (C4′), 45.48 (C3′), 53.54 (C2′), 170.22 (C1′), 175.23 (C5′); **URS part**: 15.95 (C25), 17.33 (C24), 17.63 (C26), 17.63 (C29), 19.24 (C6), 21.50 (C30), 24.15 (C27), 24.33 (C11), 24.47 (C2), 25.27 (C16), 26.81 (C3), 28.77 (C23), 29.12 (C15), 31.63 (C21), 34.03 (C7), 37.86 (C22), 38.01 (C10), 38.86 (C4), 39.33 (C1), 40.32 (C19), 40.35 (C20), 40.78 (C8), 43.18 (C14), 48.79 (C9), 49.50 (C17), 52.10 (C31), 54.36 (C18), 56.59 (C5), 126.84 (C12), 139.61 (C13), 179.76 (C28); **HRMS** (ESI) m/z: [M+H]^+^ calcd for C_36_H_58_NO_6_: 600.4264, found: 600.4267; [M-H]^-^ calcd for C_36_H_57_NO_6_Cl: 634.3874, found: 634.3887.

Compound **4c**—3β-(*l*-seryloxy)-urs-12-en-28-oic acid methyl ester hydrochloride.

Yield 75%, white solid; **Mp**: 214.6 °C (dec.); **^1^H NMR** δ [ppm]: ***l*-serine part**: 4.02 (2H, m, H3′), 4.14 (1H, m, H2′), **URS part**: 0.80 (3H, s, H26), 0.91 (3H, d, H29), 0.92 (1H, m, H5), 0.95 (3H, s, H24), 0.96 (3H, s, H23), 0.99 (3H, s, H30), 0.99 (1H, m, H20), 1.03 (3H, s, H25), 1.22 (1H, m, H1), 1.12 (1H, m, H15), 1.12 (3H, s, H27), 1.37 (1H, m, H21), 1.37 (1H, m, H7), 1.41 (1H, m, H19), 1.47 (1H, m, H6), 1.52 (1H, m, H21), 1.59 (1H, m, H6), 1.59 (1H, m, H7), 1.61 (1H, m, H22), 1.63 (1H, m, H9), 1.67 (1H, m, H22), 1.69 (1H, m, H16), 1.74 (2H, m, H2), 1.75 (1H, m, H1), 1.83 (1H, m, H15), 1.97 (2H, m, H11), 2.09 (1H, m, H16), 2.25 (1H, d, H18), 3.62 (3H, s, H31), 4.68 (1H, m, H3), 5,26 (1H, t, H12); **^13^C NMR** δ [ppm]: ***l*-serine part**: 56.20 (C2′), 60.76 (C3′), 168.81 (C1′); **URS part**: 15.95 (C25), 17.28 (C24), 17.63 (C26), 17.63 (C29), 19.25 (C6), 21.50 (C30), 24.15 (C27), 24.34 (C11), 24.54 (C2), 25.27 (C16), 28.61 (C23), 29.12 (C15), 31.63 (C21), 34.04 (C7), 37.86 (C22), 38.00 (C10), 38.96 (C4), 39.34 (C1), 40.32 (C19), 40.35 (C20), 43.18 (C8), 43.18 (C14), 48.79 (C9), 49.40 (C17), 52.10 (C31), 54.36 (C18), 56.63 (C5), 84.90 (C3), 126.86 (C12), 139.60 (C13), 179.77 (C28); **HRMS** (ESI) m/z: [M+H]^+^ calcd for C_34_H_56_NO_5_: 558.4158, found: 558.4166; [M+Na]^+^ calcd for C_34_H_55_NO_5_Na: 580.3978, found: 580.3991.

### 3.3. Biological Studies

#### 3.3.1. Cell Culture

MCF-7 and MDA-MB-231 human BC cells were purchased from the ATCC (American Type Culture Collection, Manassas, VA, USA). All cell lines were maintained in DMEM (Corning, Kennebunk, ME, USA). The medium was supplemented with 10% fetal bovine serum—FBS (Eurx, Gdansk, Poland) and 1% antimicrobial substances—penicillin/streptomycin (Corning, Kennebunk, ME, USA). The incubator asserted appropriate growth conditions, required for these cell lines: 5% of CO_2_, 37 °C, and the humidity between 90 and 95%. Cells were seeded in 100 mm round dishes; 0.05% trypsin containing 0.02% EDTA (Corning, Kennebunk, ME, USA) and phosphate-buffered saline (PBS) without calcium and magnesium (Corning, Kennebunk, ME, USA) were used to detach cells from a plate, once 80–90% cell confluence was achieved. In the next step, cells were reseeded in six-well plates (density—5 × 10^5^ of cells per well) in 1 mL of DMEM and after a 24 h incubation used in the presented tests.

#### 3.3.2. MTT Assay

To examine the effect of novel UA derivatives on cell viability, the MTT assay was performed. UA was used as a reference drug. Cells, seeded in six-well plates, were incubated for 24 h with serial dilutions (10–100 µM) of the tested compound and the reference drug in duplicates. In the next step, the liquid was aspirated above the cells and the cells were washed three times with PBS. Thereafter, 50 µL of 5 mg/mL of MTT (Sigma-Aldrich, St Louis, MO, USA) was added to 1 mL of PBS. After the required time, the MTT solution was removed and the resulting formazan crystals were dissolved in DMSO (Sigma-Aldrich, St Louis, MO, USA). The absorbance was measured using Spectrophotometer UV-VIS Helios Gamma (Unicam/ThermoFisher Scientific Inc., Waltham, MA, USA) at a wavelength of 570 nm. The absorbance obtained in untreated control cells was taken as 100%, while the survival of the cells incubated with the tested compounds was presented as a percentage of the control value [59].

#### 3.3.3. [^3^*H*]-Thymidine Incorporation Assay

The antiproliferative properties of the newly synthesized compounds were investigated using the [^3^*H*]-thymidine incorporation assay, as described in the literature [60]. Cell culture was exposed to various concentrations (10–100 µM) of novel UA derivatives and reference drug (UA) for 24 h. Thereafter, cells were washed with PBS, and 1 mL of fresh medium was added to each well. Then, 0.5 µCi of radioactive [^3^*H*]-thymidine was appended, and the incubation was continued for four hours. After the following incubation, the liquid was aspirated, and the plate was placed on ice. Cells were washed twice with 1 mL of 0.05 M Tris-HCl buffer comprising 0.11 M NaCl and then twice with 1 mL of 5% TCA acid (Stanlab, Lublin, Poland). Finally, cells were dissolved with 1 mL of 0.1 M NaOH with 1% SDS (Sigma-Aldrich, St Louis, MO, USA) at room temperature. The resulting cell lysates were transferred into scintillation vials containing 2 mL of scintillation fluid. Radioactivity was determined using Scintillation Counter 1900 TR, TRI-CARB (Packard, Perkin Elmer, Inc., San Jose, CA, USA). The intensity of DNA biosynthesis in cells was expressed in dpm of radioactive thymidine incorporated in the DNA. The radioactivity observed in untreated control cells was taken as 100%. The values of the compounds tested were expressed as a percentage of the control value.

#### 3.3.4. Determination of Bax, Caspase-7, LC3A, LC3B, Beclin-1, MMP-2, MMP-9, IL-6, and TNF-α

High-sensitivity assay kits (EIAab Science Co., Ltd., Wuhan, China; Abcam plc., Cambridge, UK) were used to determine the concentrations of selected proteins in cell lysates after 24 h incubation with a novel compound and reference drug in 1 µM and 5 µM concentrations. In brief, after trypsinization, cells were washed thrice with cold PBS and centrifuged at 1000× *g* for 5 min at 4 °C. The cells (1.5 × 10^6^) were then suspended in lysis buffer for whole cell lysates. After the second centrifugation, the cellular supernatants were frozen immediately at −70 °C. Untreated cancer cells were taken as a control. For MMP-2 and MMP-9, the concentration of proteins was measured in supernatants from cell culture. After 24 h incubation with tested compounds, the supernatants were collected and frozen at −20 °C. The microtiter plates provided in the kits were precoated with an antibody specific to the analyzed antigen. The tests were carried out according to the manufacturer’s protocols.

#### 3.3.5. Statistical Analysis

The obtained results are presented as mean ± SEM from three independent experiments performed in duplicate. Statistical analysis was performed using GraphPad Prism Version 9.3 (San Diego, CA, USA). The one-way ANOVA with Tuckey’s multiple comparison post hoc test was used to show differences between control and cancer cells exposed to varying concentrations of UA derivatives and the reference drug. Statistically significant differences were defined at * *p* ≤ 0.05.

### 3.4. Computational Details

In the computational part of this work, we used a two-step molecular docking approach with the first step similar to our previous study on diosgenin and tigogenin conjugates as antitumor agents [57], but extended it to use two target crystal structures. First, we built the models of all the molecules studied manually starting from the structure of the UA (PubChem CID: 64945) [61]. The next step was their ADME properties using the QikProp 4.6 software (Schrödinger Inc., New York, NY, USA) with default options. The most important selected ADME properties are presented in Table 3 and analyzed in Section 2.3. In the molecular docking part, we used two crystal structures (PDB codes: 2YAT and 1QKT) [62,63] of the estrogen receptor-ligand binding domain complex and the same protocol for protein preparation and molecular docking as in our previous work. In short, we used Autodock 4.2 (Scripps Research, San Diego, CA, USA). [64] with the Genetic Lamarckian Algorithm and standard options, but included 200 dockings per compound and 5,000,000 energy evaluations per docking [65]. During the docking, the following residues were treated as flexible: Glu353, Arg394, Phe404, Glu423, Phe425, and His524 (for 2YAT) and Met343, Leu346, Glu353, Leu387, Met388, Leu391, Arg394, Phe404, Met421, His524, and Leu525 (for 1QKT) while all other amino acid residues were treated as rigid. The predicted Gibbs free energies of binding and *K*_i_ values presented in Table 4 are the lowest estimates from the molecular docking of two different crystal structures of the estrogen receptor. The pKa values of the basic nitrogen-containing functional groups in compounds **3c**–**4c** and **5a**–**12a** were estimated using Epik software (Schrödinger Inc., New York, NY, USA) [66]. For comparison, we also performed local docking of the ligands found in the crystal structures 2YAT and 1QKT, but, in the former case, without the Eu^2+^ cation, for which no docking parameters are available.

## 4. Conclusions

A group of UA derivatives substituted with various amino acids and dipeptides at the C-3 position was synthesized. The compounds have been evaluated for their antiproliferative and anti-inflammatory activity. MDA-MB-231 breast cancer cells turned out to be more sensitive to most of the tested compounds. The serine (**4c**), proline (**7a**), and alanyl-isoleucine (**10a**) derivatives showed the highest cytotoxic activity against MDA-MB-231 in the MTT assay. The apoptosis pathway associated with an increase in Bax protein and caspase-7 enzyme levels was postulated as a possible mechanism of action for **4c** and **10a**. Only derivative **7a** was able to increase the release of autophagic markers (LC3A, LC3B) inducing the autophagy pathway. All tested compounds **4c**, **7a,** and **10a** decreased the concentrations of MMP-9 and MMP-2 in MDA-MB-231 cells. Compound **7a** was most effective in reducing MMP-2 levels. The in vitro anti-inflammatory efficacy of **4a**, **7a**, and **10a** was also tested in MDA-MB-231 cells. Compound **7a** showed inhibition of two proinflammatory cytokines (TNF-α and IL-6), while compound **4c** only reduced the concentration of TNF-α.

Theoretical calculations of UA derivatives were performed to predict their physical properties and biological potency. Molecular docking studies demonstrated the high binding affinity of UA derivatives to the active site of the estrogen receptor, with estimated values higher than those of the reference estradiol. Two potential modes of ligand binding were identified with details of interactions stabilizing conformers at the active site of the estrogen receptor. The ADME data showed no potential pharmacokinetic or pharmacological issues, except for high lipophilicity.

The results of our experimental and computational studies related to in vitro anticancer activity, molecular docking, and drug-like properties have shown that a new group of ursolic acid derivatives, especially compound **7a**, has an interesting profile against triple-negative breast cancer cells, i.e., cancer with poor survival prognosis.

## Data Availability

The datasets presented in the current study are available from the corresponding author on reasonable request.

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
