# Peer review of "Synthesis, Biological Activity, ADME and Molecular Docking Studies of Novel Ursolic Acid Derivatives as Potent Anticancer Agents"

_ijms, 2023, doi:10.3390/ijms24108875_

Round 1

Reviewer 1 Report

Dear Authors,

Please see detailed comments as outlined below.

Overall

All sentences must contain Include reference to literature for all given sentences. Revise throughout.

Abstract

Please check for minor English and grammar errors.

Introduction

Please ensure that each sentence contains a reference to literature. Correct throughout.

Please expand on the statistics of breast cancer to elaborate on its research problem. Eg. Countries that display the highest number of women affected by breast cancer. The rationale behind this study is brief and must be expanded on.

List these severe side effects- line 46.

Elaborate on the role of “certain enzymes” on the receptors of cancer cells in breast cancer- lines 48-50.

Please list the names of common medicinal plants that contain “UA”- line 55.

Results and Discussion

Which crude products? Please list and clarify- line 166.

Lack of discussion and reference to literature for 2.1 Chemistry and 2.2.1. Cell viability and proliferation in MCF–7 and MDA–MB–231 breast cancer cells. Please include.

Label graphs Figure: (A); (B) and avoid referring to the left and right graphs, rather state Fig. 4 A or B.

Materials and methods

What concentrations of the tested compounds were used for the MTT assays- 744.

What concentrations of the novel UA were used for the [3. H]–thymidine incorporation assay – 757

Please check for minor English and grammar errors.

Author Response

Answers for the comments of Reviewer 1:

Comments:

1) All sentences must contain Include reference to literature for all given sentences. Revise throughout.

Answer 1:

We corrected the reference according to Reviewer’s suggestion.

2)Abstract - Please check for minor English and grammar errors.

Answer 2:

We improved the minor English and grammar errors in Abstract.

3)Introduction - Please ensure that each sentence contains a reference to literature. Correct throughout

Answer 3:

We verified that the manuscript contains references to literature throughout the publication.  

4)Introduction - Please expand on the statistics of breast cancer to elaborate on its research problem. Eg. Countries that display the highest number of women affected by breast cancer. The rationale behind this study is brief and must be expanded on.

Answer 4:

According to the comment, the following paragraph have been added (doi in brackets were placed only for reviewers convenience and they do not exist in the real manuscript):

Over 2.3 million new cases and 685,000 deaths from breast cancer occurred in 2020 with geographic variation across countries and world regions. The incidence rates ranging from <40 per 100,000 females in some Asian and African countries, to over 80 per 100,000 in Australia/New Zealand, Northern America, and parts of Europe [2] (https://doi.org/10.1016/j.breast.2022.08.010). In Poland, in the years 2000-2019, breast cancer was diagnosed in 315,278 patients (the ratio of men to women was 1/100) [3]. (doi: 10.1007/s10549-022-06828-5).

5)Introduction List these severe side effects- line 46.

Answer 5:

According to the comment the following paragraph have been added:

…resulting in severe side effects, such as secondary (therapeutic-induced) malignancies, nephro-, hepato-, neuro-, cardio-, and ototoxicity [5] (https://doi.org/10.3390/cancers14030627)

6)Introduction Elaborate on the role of “certain enzymes” on the receptors of cancer cells in breast cancer- lines 48-50.

Answer 6

According to the comment the role of ‘’certain enzymes’’ have been listed:

…activity of certain enzymes or  overexpression of receptors located on the surface of tumor cells [6] (doi: 10.1007/s00277-013-1947-2). For example, overexpression of CES2, belonging to the largest class of enzymes in humans called serine hydrolases (SH), has been observed in pancreatic adenocarcinoma compared to paired non-cancerous tissues [7] (doi: 10.1093/jnci/djv132)

7) Introduction Please list the names of common medicinal plants that contain “UA”- line 55.

Answer 7

The names of common medicinal plants has been added in the Introduction.

UA is a pentacyclic triterpenoid derived from berries, leaves, flowers, and fruits of medicinal plants (e.g. lavender, oregano, thyme, marigold [8] https://doi.org/10.3390/ijms22094599(https://doi.org/10.3390/ijms22094599)

8) Results and Discussion - Which crude products? Please list and clarify- line 166.

Answer 8

Adding (lines 175-176) the paragraph below we have explained what crude products means.

The crude products of coupling reactions were then purified by flash chromatography (ethyl acetate/hexane) to yield (46 to 81%) 39 intermediates

9) Results and Discussion -  Lack of discussion and reference to literature for 2.1 Chemistry and 2.2.1. Cell viability and proliferation in MCF–7 and MDA–MB–231 breast cancer cells. Please include

Answer 7

The reference to the literature have been added according to the Reviewer's comment.

10) Results and Discussion - Label graphs Figure: (A); (B) and avoid referring to the left and right graphs, rather state Fig. 4 A or B.

Answer 10

The Figure has been corrected according to the Reviewer's comment.

11) Materials and methods - What concentrations of the tested compounds were used for the MTT assays- 744.

Answer 11

The concentrations of the tested compounds were 10-100 µM. This value has been  included in the publication.

12) Materials and methods - What concentrations of the novel UA were used for the [3. H]–thymidine incorporation assay – 757

Answer 12

The concentrations of the tested compounds were 10-100 µM. This value has been included in the publication.

13) Please check for minor English and grammar errors.

We corrected the minor English and grammar errors in the publication.

Reviewer 2 Report

In the manuscript “ Synthesis, biological activity, ADME and molecular docking studies of novel ursolic acid derivatives as potent anticancer agents”, the Authors synthesized a set of ursolic acid derivatives substituted with various amino acids and dipeptides at C–3 position of the steroid skeleton. The effect of new ursolic acid derivatives on the cell viability, proliferation,  apoptosis, metalloproteinase–2 and –9 activity, autophagic activity, and  inflammatory status of human breast cancer cells (MCF–7 and MDA–MB–231 cell lines) was studied (determination of Bax, caspase–7, LC3A, LC3B, Beclin–1, MMP–2, MMP–9, IL–6, and TNF–α). For all synthesized compounds their ADME properties were predicted, and molecular docking to estrogen receptor was performed. The results showed that ursolic acid analogs displayed antiproliferative and anti-inflammatory activity  in human breast cancer cells.

The manuscript is interesting and well written (Introduction, Materials and methods, Results and Discussion, Conclusions), and the results are clearly presented.

Author Response

Answers for the comments of Reviewer 2:

Thank you very much for your careful Review. We are very grateful for your positive comments.

Reviewer 3 Report

In the experimental section, the number of the compounds should be expressed in bold for uniformity.

Author Response

Answers for the comments of Reviewer 3:

Thank you very much for your Review. We are very grateful for your positive comments.

Comment:

1)In the experimental section, the number of the compounds should be expressed in bold for uniformity

Answer 1

The numbers  have been bolded in the Experimental Section.

Reviewer 4 Report

Journal of International Journal of Molecular Sciences

Research Article;

The article entitled “Synthesis, biological activity, ADME and molecular docking studies of novel ursolic acid derivatives as potent anticancer agents’’. The author investigated the A series of novel ursolic acid derivatives substituted with various amino acids (AAs) or dipeptides (DP) at C–3 position of the steroid skeleton. The compounds have been prepared by an esterification of ursolic acid with the corresponding amino acids. The cytotoxic activity of the synthetized conjugates was determined using the breast cancer cell line showed that derivatives (L–seryloxy–, L–prolyloxy– and L–alanyl–L–isoleucyloxy–) displayed low micromolar IC50 values as well as decreased the concentrations of matrix metalloproteinases 2 and 9. The study also performed the antiproliferative assay via activating caspase–7 and the proapoptotic protein Bax, through the apoptosis pathway. L–prolyloxy– derivative showed induced the autophagy process. This ursolic acid derivative also showed inhibition of pro-inflammatory cytokines TNF–α and IL–6.

I carefully read the manuscript and it needs some critical revision for publication in the journal. There are some common mistakes, references and English language problem in the article which should be corrected by the authors. After the correction of all the mistakes, I suggestion to include in vivo data for more confirmation the article could be considered for publication in the prestigious International Journal of Molecular Sciences Journal.

Comments for Authors

Ø  The author needs to revise the introduction section. Introduction section is too long. The author needs to revise and make it short and meaningful.

Ø  In Figure 1. The author mentioned the structure of ursolic acid and explained the activities of ursolic acid. The reference 23 (line number 925-926) didn’t describe such activities which needs critically revision. The figure 1 didn’t matched with research article protocol.

Ø  The author mentioned the name of the (author et al) during whole manuscript. Once you mentioned the reference there is no need to mentioned the (author et al) of any reference

Ø  Write keywords in alphabetical order.

Ø  The author needs to revise the subtitle of Result and Discussion and Materials and methods sections.

Ø  The author needs to revise conclusion, it should be no more the single paragraph of about 4 to 5 lines. There no need to explain the result section in conclusion.

Cite the following references;

v  DOI: 10.2174/1871520622666220831124321

v  DOI: 10.1038/s41419-021-03771-z

Author Response

Answers for the comments of Reviewer 4:

Comments:
1)There are some common mistakes, references and English language problem in the article which should be corrected by the authors.

Answer 1

The manuscript has been revised according to the Reviewer's comment.

2) I suggestion to include in vivo data for more confirmation the article could be considered for publication in the prestigious International Journal of Molecular Sciences Journal

Answer 2

This project aimed at the synthesis of new ursolic acid derivatives and their in vitro biological evaluation. These preliminary studies only allowed us to select a candidates for further in vivo studies. In the future project, we plan to perform in vivo research on the  selected compounds, as the Reviewer rightly suggested.

3) The author needs to revise the introduction section. Introduction section is too long. The author needs to revise and make it short and meaningful.

Answer 3

The introduction has been revised according to the Reviewer's comment but was also supplemented with additional information according to the comments of another Reviewer.

4)In Figure 1. The author mentioned the structure of ursolic acid and explained the activities of ursolic acid. The reference 23 (line number 925-926) didn’t describe such activities which needs critically revision. The figure 1 didn’t matched with research article protocol.

Answer 4

The Figure 1 has been corrected according to the Reviewer's comment.

5)The author mentioned the name of the (author et al) during whole manuscript. Once you mentioned the reference there is no need to mentioned the (author et al) of any reference.

Answer 5

This notation is commonly used in scientific papers  and has been accepted in our publications by the Editors of different scientific journals. The suggested correction requires changing the form of writing to the passive voice and was not recommended by other Reviewers.

6)Write keywords in alphabetical order.

Answer 6

The keywords have been written in alphabetical order.

7)The author needs to revise the subtitle of Result and Discussion and Materials and methods sections.

Answer 7

We corrected the subtitles according to Reviewer’s suggestion.

8)The author needs to revise conclusion, it should be no more the single paragraph of about 4 to 5 lines. There no need to explain the result section in conclusion.

Answer 8

We have improved and shortened the Conclusion according to Reviewer’s suggestion.

Round 2

Reviewer 4 Report

the author didn't revise the manuscript accordingly and just answering according to other reviewers.  there are many mistakes which could be critically revised with bird eye. I don't think that this manuscript is able to publish in prestigious journal.